# A numerical framework for mechano-regulated tendon healing—Simulation of early regeneration of the Achilles tendon

**Thomas Notermans**[1]*, **Petri Tanska**[2], **Rami K. Korhonen**[2], **Hanifeh Khayyeri**[1], **Hanna Isaksson**[1]

1 Department of Biomedical Engineering, Lund University, Lund, Sweden, 2 Department of Applied Physics, University of Eastern Finland, Kuopio, Finland

* thomas.notermans@bme.lth.se

**Data Availability Statement:** All relevant data are within the manuscript and its Supporting Information files.

## Abstract

Mechano-regulation during tendon healing, i.e. the relationship between mechanical stimuli and cellular response, has received more attention recently. However, the basic mechano-biological mechanisms governing tendon healing after a rupture are still not well-understood. Literature has reported spatial and temporal variations in the healing of ruptured tendon tissue. In this study, we explored a computational modeling approach to describe tendon healing. In particular, a novel 3D mechano-regulatory framework was developed to investigate spatio-temporal evolution of collagen content and orientation, and temporal evolution of tendon stiffness during early tendon healing. Based on an extensive literature search, two possible relationships were proposed to connect levels of mechanical stimuli to collagen production. Since literature remains unclear on strain-dependent collagen production at high levels of strain, the two investigated production laws explored the presence or absence of collagen production upon non-physiologically high levels of strain (>15%). Implementation in a finite element framework, pointed to large spatial variations in strain magnitudes within the callus tissue, which resulted in predictions of distinct spatial distributions of collagen over time. The simulations showed that the magnitude of strain was highest in the tendon core along the central axis, and decreased towards the outer periphery. Consequently, decreased levels of collagen production for high levels of tensile strain were shown to accurately predict the experimentally observed delayed collagen production in the tendon core. In addition, our healing framework predicted evolution of collagen orientation towards alignment with the tendon axis and the overall predicted tendon stiffness agreed well with experimental data. In this study, we explored the capability of a numerical model to describe spatial and temporal variations in tendon healing and we identified that understanding mechano-regulated collagen production can play a key role in explaining heterogeneities observed during tendon healing.

**Funding:** HI received funding from Knut and Alice Wallenberg KAW Foundation (https://kaw.wallenberg.org/en) (Wallenberg Academy Fellows 2017.0221). RK and HI received funding from European Union's Horizon 2020 research and innovation programme under the Marie Sklodowska-Curie grant agreement no. 713645. The funders had no role in study design, data collection and analysis, decision to publish, or preparation of the manuscript.

**Competing interests:** The authors have declared that no competing interests exist.

## Author summary

The frequency of Achilles tendon ruptures has increased over the last decades. Treatments can involve different loading or unloading strategies of the Achilles tendon during healing. However, there is no consensus on the optimal treatment since the effect of loading on Achilles tendon healing is not fully understood. Recent experimental studies have shown that the tendon heals differently in different regions, particularly the core of the tendon callus seems to behave differently from the outside of the tendon. To better understand these spatial variations in Achilles tendon healing, a 3D computational model of a normally loaded healing tendon was created. We predicted tissue formation and reorganization upon mechanical stimulation of the healing tissue. We observed that strain-dependent tissue formation could explain recent observations of decreased tissue formation in the core of the healing callus during early tendon healing. This work investigates how mechanical stimuli affect the formation and reorganization of the newly formed tissue, considering both distributions in space and over time.

## Introduction

The biomechanical function of tendons is primarily a result of its unique tissue composition, organization and mechanical properties. The innate healing process following a tendon rupture has a limited capability of restoring intact tendon properties [1,2]. Tendon healing is initiated with an acute inflammatory stage, which is followed by production and alignment of extracellular matrix (ECM; Collagen type-I and III) to restore the loadbearing capacity. Studies that investigated collagen formation and orientation during Achilles tendon healing described an increasing alignment of collagen with time in rats [3] and rabbits [4,5]. Interestingly, Sasaki et al., also revealed spatial variations in the collagen organization during the first 4 weeks of healing, where they observed clear differences in organization between the core (along the central axis of the tendon) and the periphery (towards the outer edge of the tendon along the paratenon) of the tendon [3]. It is widely accepted that mechanical loading affects tendon healing. Numerous experimental studies have investigated the temporal changes in collagen content, collagen organization and mechanical properties during tendon healing comparing how different degrees of mechanical loading affect tendon healing [6–10]. However, extensive experimental data describing the spatio-temporal evolution of collagen properties (e.g. content and orientation) in the callus during tendon healing has been limited.

Our recent study described the spatio-temporal evolution of collagen properties during tendon healing in rat Achilles tendons, and showed large spatial variation within the callus in terms of collagen structure and orientation [11]. Different structural collagen properties were measured in healing transected rat Achilles tendon, on the fibril-scale by Small-Angle X-ray Scattering (SAXS) and on the tissue-scale by histology. The data implied decreased collagen levels and delayed maturation in the center of the tendon callus (core) during the first 4 weeks of healing (Fig 1) [11]. Furthermore, histological assessment revealed distinct organized fibrous tissue at the periphery of the callus whereas at 4 weeks post-rupture well-developed collagen organization was found both in the periphery and the core of the healing callus. Other studies have shown similar results, e.g. a recent study showed a similar heterogeneous collagen pattern with thin collagen fibrils in the core and thicker collagen fibrils in the periphery at 2 weeks post tendon-rupture in Achilles tendon of male Sprague-Dawley rats [12]. Furthermore, histological analysis on flexor tendon healing in mice revealed decreased collagen staining in the core of the defect at 2 weeks post-repair [13]. These observations in combination with our

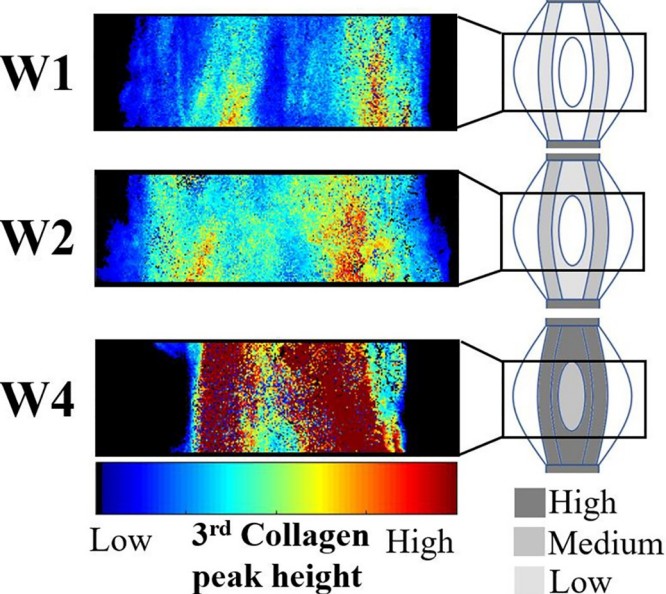

**Fig 1. Representative data from Small Angle X-ray Scattering (SAXS) experiments in healing rat Achilles tendon at 1, 2 and 4 weeks post-transection [11].** The third collagen peak height is representative of the intrafibrillar order in collagen fibrils. The data support that collagen production and maturation in the core of the tendon appears to be delayed and decreased in rats subjected to normal free cage activity.

data indicates a delayed production and maturation of collagen fibrils and fibers in the core of the healing callus during early tendon healing.

Few numerical studies have investigated mechanobiological mechanisms underlying the temporal evolution of collagen content, collagen alignment and overall tissue mechanical properties during tendon healing [14,15]. However, several experimental investigations have described heterogeneity and spatial variations in the evolution of collagen formation and organization during tendon healing [3–5,12,13,16,17]. To the authors' knowledge, no numerical study has investigated both spatial and temporal evolution of collagen content and organization during tendon healing. Spatial and temporal variations of mechanical stimuli in the regenerating tendon may contribute to heterogeneous tendon healing. To improve understanding of the mechanobiological mechanisms underlying tendon healing, we explored if mechano-regulated processes that govern tendon healing could explain both spatial and temporal variations in tendon healing.

In this study, a novel 3D computational mechanobiological framework was developed to predict mechano-regulated collagen production, reorientation and overall tissue stiffness during tendon healing in the rat Achilles tendon. In particular, we modeled healing of a full-width transection without surgical repair. We investigated the evolution of spatio-temporal collagen content using strain-dependent production laws. Literature data implies that collagen production decreases for supraphysiological (>15% strain) strain levels [18–24]. We therefore hypothesized that the absence of collagen production for supraphysiological strains (>15% strain) would be vital to predict the experimental observation of decreased collagen production in the healing tendon core.

## Methods

A representative healing tendon was implemented using finite element analysis (Abaqus v2017, Dassault Systèmes Simulia Corp., Johnston, RI, USA). Tensile loading was applied to

mimic normal loading of the healing tendon. An iterative process was implemented in Matlab (Matlab R2019b) to compute mechano-regulated collagen production and collagen reorientation in the healing callus on a day-to-day basis (Fig 2).

### Finite element model

The geometry, boundary conditions and mesh for the healing Achilles tendon are described in Fig 3. The geometry was based on the average of 10 rat Achilles tendon geometries at week 1 post-rupture from our recent experimental study [11], consisting of two intact stumps with longitudinally aligned collagen and a bulging healing callus.

An established fibre-reinforced transversely isotropic poro-visco-elastic material model was implemented in a 3D finite element framework to model the contributions of collagen, ground substance and fluid [26] to the overall mechanics of the tendon. Briefly, the material behavior of the collagen fibrils was modeled according to the viscoelastic standard linear solid (SLS) model:

$$P_s = \begin{cases} E_s(e^{K_s \varepsilon_s} - 1), \varepsilon_s > 0 \\ 0, \ \varepsilon_s \leq 0 \end{cases} \tag{1}$$

$$P_\eta = \eta \frac{d\varepsilon_\eta}{dt}, \tag{2}$$

$$P_f = P_s + P_\eta = P_s + \eta \frac{d\varepsilon_\eta}{dt}, \tag{3}$$

where $P_s$, $P_\eta$ and $P_f$ are the first Piola-Kirchhoff stress in the springs, dashpot and the whole SLS model, respectively, $E_s$ and $K_s$ are the linear and exponential modulus parameters, $\eta$ is the viscosity of the dashpot, $\varepsilon_s$ is the strain in the spring, and $\varepsilon_\eta$ is the strain in the dashpot.

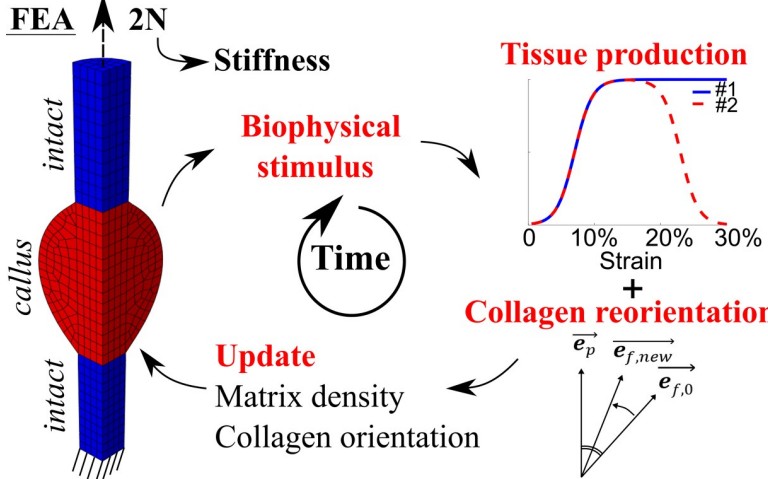

**Fig 2. Schematic overview of the numerical framework.** Finite element analysis (FEA) was used to determine local magnitudes and directions of maximum principal strain in the healing callus. Peak daily loading (maximum principal strain magnitudes) guided the heterogeneous mechano-regulated production of collagen and ECM. Reorientation was implemented such that collagen fibrils were gradually reoriented (e_f,0 ➔ e_f,new) towards the maximum principal strain direction (e_p)[25]. From the force-displacement curves during the daily external load, a tissue scale stiffness of the whole tendon was determined at peak force (2N).

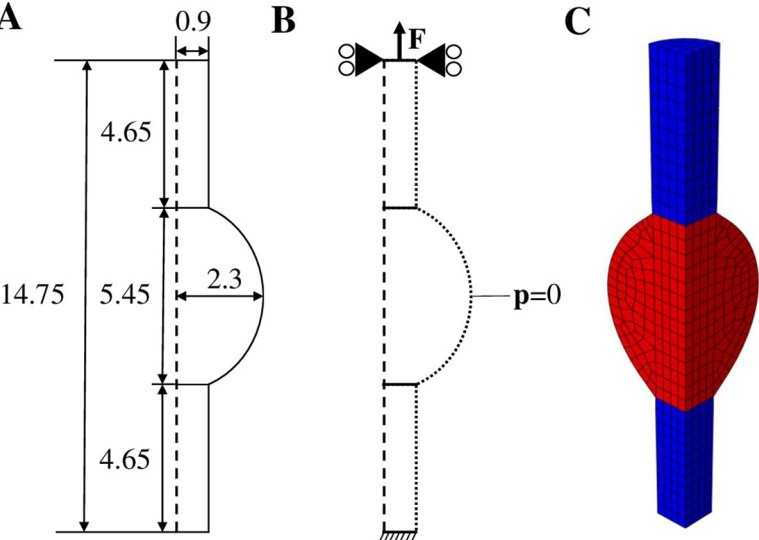

**Fig 3. Geometry, boundary conditions and mesh.** Longitudinal cross-section of the implemented geometry describing the dimensions (mm) of the intact stumps and healing callus (A) and the boundary conditions (B), where a force is applied to the top surface, the top surface nodes are only allowed to move in the longitudinal direction, the bottom surface nodes are clamped, and zero pore pressure is prescribed to the nodes on the outer surface to allow free fluid flow. The implemented finite element mesh was a quarter of a cylinder, consisting of two intact stumps (denoted in blue) and the healing callus (denoted in red). Using symmetry conditions, this mimicked a complete 3D cylinder.

Subsequently, the Cauchy stress per collagen fibril is defined according:

$$\boldsymbol{\sigma}_f = \frac{\lambda}{J} P_f \overrightarrow{\boldsymbol{e}_f} \overrightarrow{\boldsymbol{e}_f}^{\mathrm{T}}, \tag{4}$$

where $\lambda = \frac{\|\overrightarrow{\boldsymbol{e}_f}\|}{\|\boldsymbol{e}_{f,0}\|}$ is the fibril stretch, $\overrightarrow{\boldsymbol{e}_f}$ and $\overrightarrow{\boldsymbol{e}_{f,0}}$ are unit vectors describing the current and initial fibril direction respectively. The non-fibrillar ground substance was modeled by an orthotropic strain-energy formulation:

$$W_{ort} = \frac{1}{2}\sum_{i,j}^{3} a_{ij}\mathrm{tr}(\boldsymbol{E} \cdot \boldsymbol{L}_{ii})\mathrm{tr}(\boldsymbol{E} \cdot \boldsymbol{L}_{jj}) + \sum_{i,j\neq 1}^{3} G_{ij}\mathrm{tr}(\boldsymbol{E} \cdot \boldsymbol{L}_{ii} \cdot \boldsymbol{E} \cdot \boldsymbol{L}_{jj}), \tag{5}$$

that relates the stiffness parameters ($a_{ij}$, $G_{ij}$) to the Green-Lagrange strain tensor ($\boldsymbol{E}$) using an orthogonal base formulation where $\boldsymbol{L}_{ii}$ represents a dyadic product of a set of orthogonal unit base vectors. This material definition allows the ground substance to have different material properties (modulus and Poisson's ratio) in the longitudinal (2-direction) and transverse plane (1- and 3- direction). The Cauchy stress for the non-fibrillar ground substance was calculated according:

$$\boldsymbol{\sigma}_m = \frac{1}{J}\boldsymbol{F} \cdot \frac{\partial W}{\partial \boldsymbol{E}} \cdot \boldsymbol{F}^{\mathrm{T}}, \tag{6}$$

with deformation gradient ($\boldsymbol{F}$) and jacobian ($J = \det(\boldsymbol{F})$). We modeled strain-dependent fluid flow using a poroelastic formulation following Darcy's law with a void ratio dependent permeability of the tendon according to (additional details in [26]):

$$k = k_0 \left(\frac{1+e}{1+e_0}\right)^{M_k}, \tag{7}$$

where $k$ describes the permeability, $k_0$ the initial permeability, $e_0$ and $e$ the initial and current void ratio and $M_k$ is a parameter that describes the nonlinear relationship between the permeability and void-ratio.

Intact constitutive tendon properties were used for the intact tendon stumps [27,28]. For the healing callus, a density function $\rho$ was introduced to scale the Cauchy stress exerted by the collagen and ground substance in intact tendon to callus properties according to:

$$\sigma_{\text{callus}}^{\text{collagen}} = \rho * \sigma_{\text{intact}}^{\text{collagen}},$$ (8)

$$\sigma_{\text{callus}}^{\text{ground substance}} = \rho * \sigma_{\text{intact}}^{\text{ground substance}},$$ (9)

An initial callus density (collagen and ground substance) of 1% w.r.t. intact tendon was implemented to match the day-3 tendon stiffness reported in Eliasson et al. (2009), and the initial fibrils were organized randomly in 13 directions in every material point [29,30].

The daily mechanical load consisted of a preload of 0.1 N (loading rate: 0.1 N/s), followed by an external tensile load (loading rate: 1.1 N/s). The applied load magnitude increased from an initial 0.25 N at day 1 post-rupture to the assumed physiological load of 2 N [31,32] at 5 days post-rupture, and was then kept at 2 N for the remaining time.

## Adaptive mechanobiological model

Using strain-regulated production laws, callus matrix (collagen and ground substance) was produced depending on the magnitude of the maximum principal strain. A comprehensive literature study of in vitro studies investigating strain magnitude-dependent collagen production was used to propose two different tensile strain-dependent collagen production laws (Fig 4).

The first production law is implemented similarly as in Chen et al. (2018) and Richardson et al. (2018) and describes an initial increase in strain-dependent collagen production with increasing strain levels, followed by a constant level of collagen production for strains exceeding ~10% (Fig 4). Literature reports a wide range of strain-dependent collagen production for strains above 10% [18–24,33–36]. Some studies reported that gene expression and/or protein production of collagen 1 and/or 3 are down-regulated [21,24,35], at baseline level [19,24,35], or only mildly up-regulated [24,35] with respect to non-mechanically stimulated fibroblasts. In addition, *in vitro* studies investigating multiple magnitudes of cyclic strain stimulation display some 'optimal' levels around 10% tissue-level strain for inducing collagen synthesis which when exceeded result in decreased collagen gene expression and/or production [23,24,35]. Considering these experimental findings, we propose a second production law that entails both the established initial increase of production with an increase in strain magnitude (as seen in production law 1), followed by a decrease of collagen production upon 'overstimulation' (Fig 4). The strain-regulated production laws were defined similarly to Richardson et al. (2018) according to:

$$\text{Production Factor} = \frac{1}{1 + e^{-k_{\text{sig}}(\text{strain}-\text{C1})}} \text{ for strain < transition point}$$

$$\text{Production Factor} = 1 - \frac{1}{1 + e^{-k_{\text{sig}}(\text{strain}-\text{C2})}} \text{ for strain > transition point}$$

leading to the following production laws (Fig 4), where C1, C2 and $k_{\text{sig}}$ are shape parameters.

Previous studies on collagen production during rat and mouse tendon healing suggest that the bulk of collagen production occurs in the first 4 to 6 weeks of healing [6,43–51], implying a daily production rate of 2–4% a day. Based on this, a maximum amount of collagen produced per day

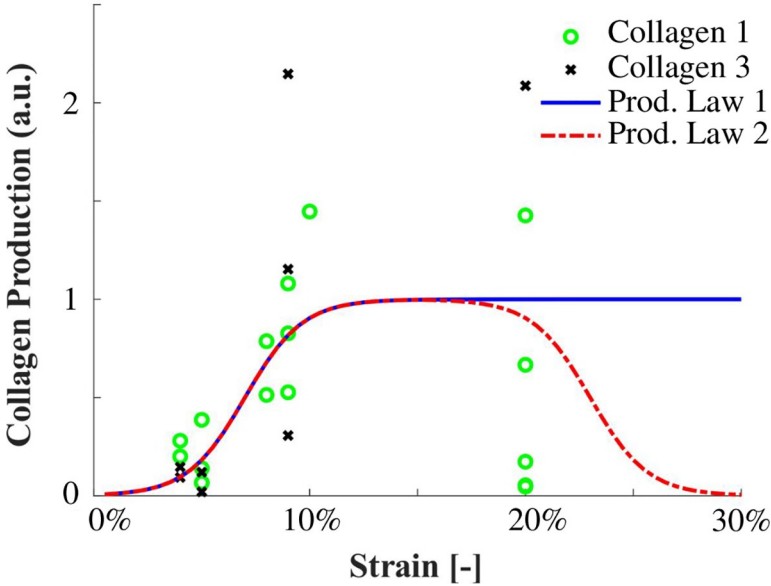

**Fig 4. Strain-dependent collagen production laws determined from literature ($k_{sig}$ = 75; transition point = 0.15; C1 = 0.07; C2 = 0.03).** Data points in green circles or black dots are collected from: [18,22,34,37–42], see S1 Table.

of 2% of the intact collagen content was assumed. Additionally, in order to captures inflammatory-driven collagen production during the first days of acute inflammation, special conditions were applied during the first 4 days of healing. To model this, mechano-regulated production of maximum 1%/day was combined with an additional constant inflammatory-driven baseline production rate of 1%/day [4]. Subsequently, from day 5 to day 28 the daily production rate was purely dependent on mechano-regulated production (up to 2%/day). We assumed that the ground substance, i.e. non-collagenous matrix, is produced at a similar rate as collagen, thus the spatio-temporal evolution of collagen and ground substance occur in a similar fashion.

## Collagen reorientation

The orientation of the collagen fibrils in the healing callus was initially assumed to be random with 13 different directions [29,30]. The fibrils were then allowed to gradually re-orient towards the local maximum principal strain direction, as earlier implemented for articular cartilage [25] according to:

$$\overrightarrow{e_{f,new}} = \exp(\kappa \alpha \boldsymbol{R})\overrightarrow{e_{f,0}} \tag{10}$$

where $\alpha$ is the angle between the current fibril direction ($\overrightarrow{e_{f,0}}$) and the direction of maximum principal strain, rate parameter $\kappa$, new fibril orientation ($\overrightarrow{e_{f,new}}$) and a rotation matrix ($\boldsymbol{R}$) that rotates the fibril towards the direction of maximum principal strain. The reorientation rate in the model ($\kappa$ = 0.06) was adapted to capture the literature finding that the bulk of longitudinal alignment is completed at 4 weeks post-rupture [3,52] (S4 and S11 Figs). A well-established alignment criterion [14,53,54] was used to quantify the alignment of the collagen fibrils with the direction of principal strain:

$$S = \frac{1}{N}\sum_{i=1}^{N}\cos(2\alpha) \tag{11}$$

where N is the number of collagen fibrils in the healing callus. The degree of alignment (S) varies from -1 (perpendicular to the maximum principal strain) to 1 (aligned with the maximum principal strain).

## Analysis

Throughout the healing process, the spatio-temporal evolution of collagen content and collagen orientation were the primary output. In addition, the daily tensile load was also used to measure the temporal evolution of the overall tendon stiffness. Spatial collagen distribution and the overall tendon stiffness at 1, 2 and 4 weeks was qualitatively compared with our previous study [11]. To explore the effects of our modeling assumptions we performed sensitivity analysis on the mesh density, mesh geometry, loading rate, reorientation rate, exact definition of production law 2, and long-term predictions.

## Results

The spatio-temporal evolution of tensile strain in the healing callus for production law 1 and 2 is displayed in Fig 5. The magnitude of tensile strain was predicted to be high in the core of the tendon callus between the tendon stumps in the first 2 weeks post-rupture. Particularly the area around the stump interfaces displayed high magnitudes of tensile strain. Within 4 weeks, tensile strain returned to the physiological range of the respective stimuli. A mesh convergence study showed that an increasing mesh density was not found to significantly affect the accuracy of the predicted strain maps.

With production law 1, the tensile strain stimulus predicted high collagen production at the stump interface and in-between the stumps, bridging the gap between the two stumps (Fig 6). Production law 1 predicts most collagen production in the tendon core at all times. Interestingly, production law 2 predicted reduced collagen content (64%, 35%, 23% reduction w.r.t. production law 1 at week 1, 2 and 4 respectively) in the core (Fig 6). Instead, high peripheral collagen production was observed for the first 2 weeks of healing. This was followed later by higher collagen production in-between the stumps. Production law 1 predicted a strong development of collagen content at the stump-callus interface, whereas this production of collagen

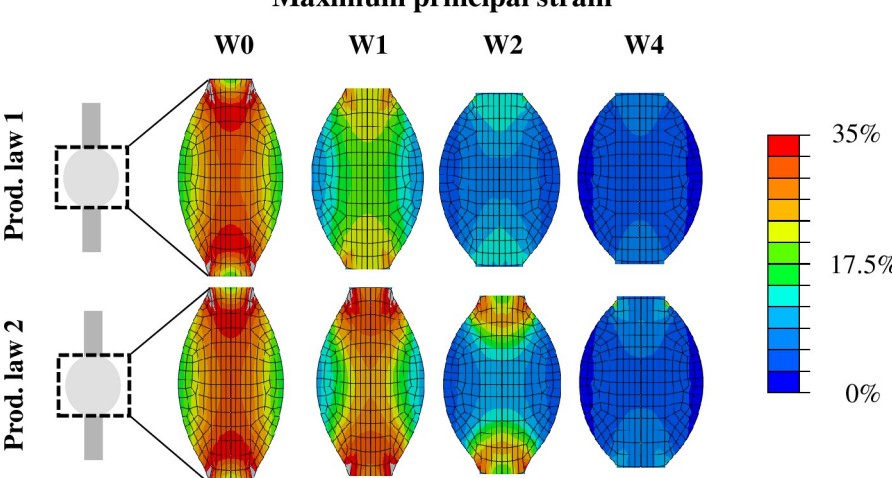

**Fig 5. Spatio-temporal evolution of the magnitude of tensile strain that locally governed collagen production through production law 1 and 2.** The deformed geometry at maximum loading is displayed. The strain maps are presented for 0, 1, 2 and 4 weeks post-rupture.

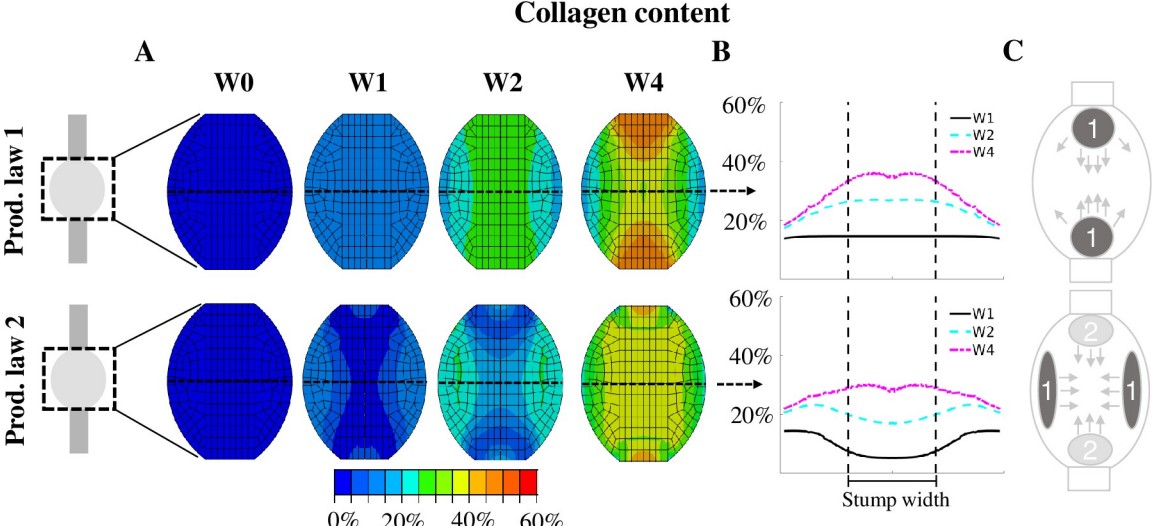

**Fig 6. Spatio-temporal evolution of collagen content predicted with production law 1 and 2.** The collagen content is displayed for a longitudinal section of the healing tendon callus (A) and across the mid-tendon cross-section (B) at 0, 1, 2 and 4 weeks post-rupture. The collagen content is normalized to the intact tendon. Additionally, a schematic description of the spatio-temporal evolution is displayed (C). The dark grey spots (1) denote the initial location of collagen production, whereas the light grey areas (2) denote the secondary location of major collagen production that arise after 2 or 3 weeks of tendon healing. The arrows denote how this production spreads throughout the callus.

was absent for production law 2 during the first 2 weeks of healing. Sensitivity analysis of the exact definition of production law 2 (S1 and S2 Figs), the reorientation rate (S5 Fig), mesh geometry (S6–S8 Figs), mesh density (minor effect) and loading rate (minor effect) showed that nearly all perturbations still predicted increased collagen production in the callus periphery at week 1. The effects were negligible or minor. The geometry of the callus affected collagen production the most, i.e. increasing the callus overlap over the intact tendon stumps resulted in a more homogeneous collagen distribution at week 1 (S8 Fig). A simulation of long-term healing with production law 2 showed that the evolution of collagen content stagnates and the value changes less than 5% after 6 weeks of healing (S11 Fig).

The spatio-temporal evolution of collagen orientation was similar for production law 1 and 2 (Fig 7). The collagen orientation displayed an overall alignment with the longitudinal axis over time where the bulk of collagen alignment was completed within 4 weeks of tendon healing. The corners of the callus (top & bottom, left & right) displayed more diagonal fibril orientations (Fig 7). A simulation of long-term healing with production law 2 showed that the evolution of alignment stagnates and the value changes less than 5% after 4 weeks of healing (S11 Fig).

Both production laws predicted temporal stiffness evolution that was within 1 standard deviation of the experimental stiffness (Fig 8). Particularly the stiffness evolution predicted by production law 2 corresponded very well to the experimental data at 1 and 2 weeks of healing. Production law 1 predicted higher tendon stiffness than production law 2, however, the difference between the two models decreased over time. The temporal evolution of stiffness predicted by both production laws described a stiffness recovery that mainly occurred during the first 4 weeks of healing. In our sensitivity analysis of production law 2 (S3 Fig), reorientation rate (S5 Fig), loading rate (minor effect) all simulations still predicted stiffness evolution within 1 standard deviation of the experimental data. A simulation of long-term healing with

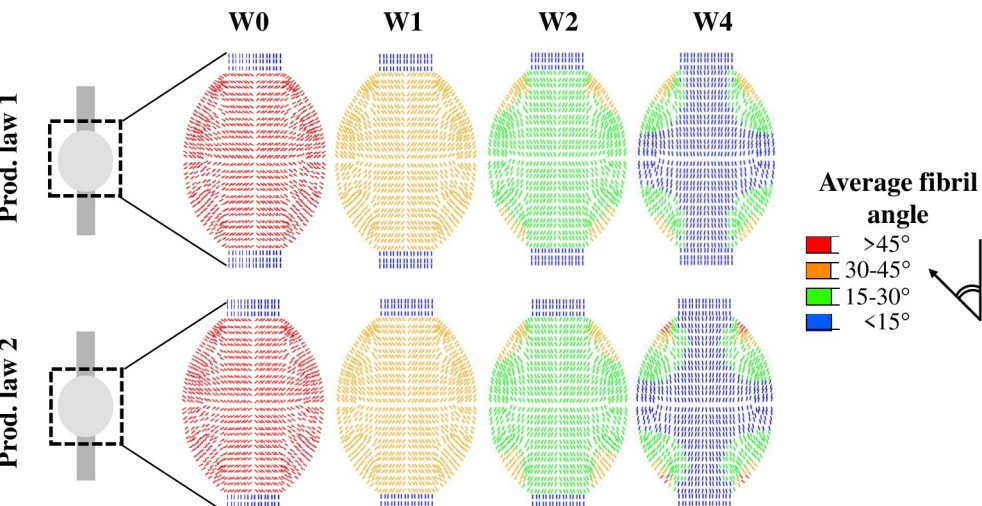

**Fig 7. Spatio-temporal evolution of fibril reorientation towards longitudinal alignment predicted with production law 1 and 2.** The average angle between the collagen fibrils (13 fibrils/material point) and the longitudinal axis is shown for 0, 1, 2 and 4 weeks post-rupture.

production law 2 showed that the stiffness evolution stagnates and the value changes less than 5% after 5 weeks of healing (S11 Fig).

## Discussion

In this study, we have explored a numerical framework to describe spatial and temporal variations observed in tendon healing. In particular, we proposed a mechano-regulatory framework to describe spatio-temporal evolution of collagen content and alignment and temporal evolution of overall tissue stiffness during tendon healing. Every iteration (representing a day of healing), a mechanical load was applied to the tendon to determine the spatial distribution of maximum principal strain magnitude and direction. On a daily basis, the maximum principal strain magnitude and direction govern tissue collagen production and reorientation, respectively. With increasing collagen production and longitudinal collagen alignment, the overall tendon stiffness increases and thus strain levels decrease. As strain levels decrease, tissue production will decrease. A newly proposed tensile strain-dependent production law (law 2) predicted recently found experimental observations that collagen production in the healing tendon core is decreased and delayed during the first 2 weeks of tendon healing in transected rat Achilles tendons during free cage activity. The explored models imply that decreased collagen production upon high loading levels ('overloading') may explain the observed lower levels of collagen content in the tendon core during tendon healing.

For the first time, the spatial and temporal evolution of callus deformation during early tendon healing was investigated. The strain distribution predicted by our healing framework (Fig 5) revealed supraphysiological levels of strain (>20%) during the first days of healing throughout the whole callus. In comparison, other numerical healing frameworks only considered strains up to 10% during tendon healing. In particular, musculoskeletal modeling of the rat hindlimb estimated a maximum tensile strain of 4.3% or 7.5% unrepaired and repaired tendons experiencing full loading [14]. In another healing study, strain magnitudes up to 10% were investigated [15]. Furthermore, our finite element simulations revealed a high spatial variation in strain levels. Particularly the stump-callus interfaces and the central longitudinal axis displayed the highest levels of strain throughout the simulated 4 weeks of healing. These results

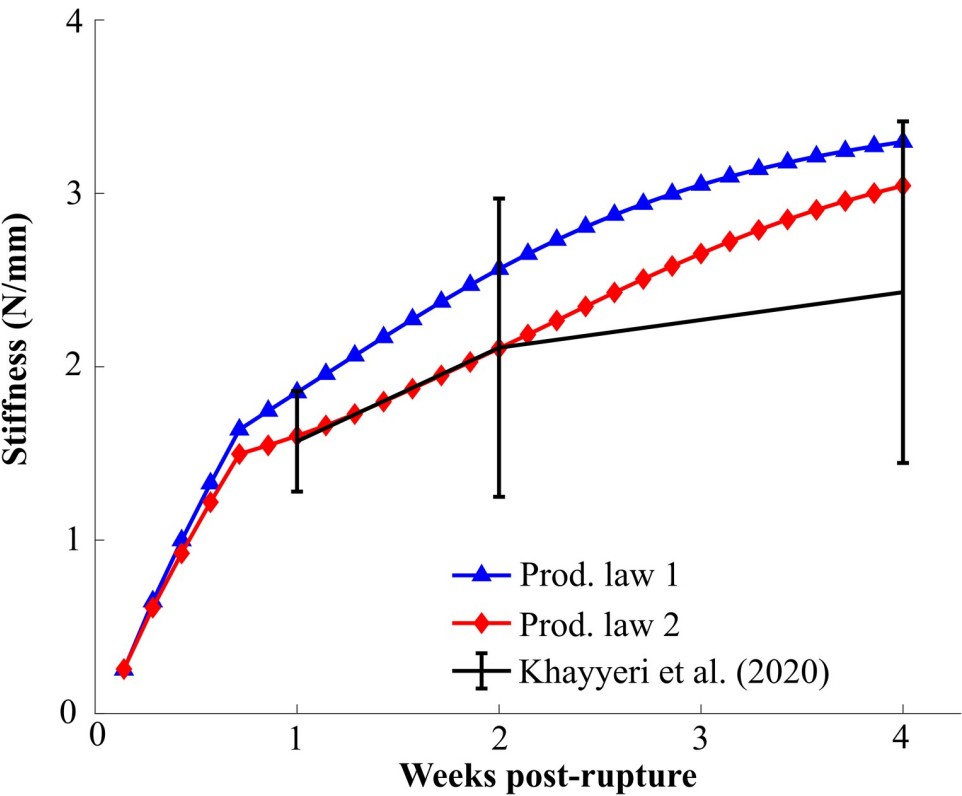

**Fig 8. Evolution of the predicted model stiffness (N/mm) for both production laws.** Khayyeri et al. (2020) reported stiffness at 1, 2 and 4 weeks post-rupture (mean±standard deviation). All stiffnesses were determined at a 2N load.

imply that mechano-dependent processes could play a larger role at the stump-callus interfaces and in the core of healing tendon. On the other hand, the peripheral region of the tendon callus appears to be subjected to more moderate levels of strain stimulation. This may stimulate the healing process in the periphery of the callus due to a more anabolic state induced by more physiological levels of mechanical stimulation, as explored by strain-dependent production law 2.

Our strain-regulated collagen production laws predicted that increased levels of tensile strain may cause delayed collagen production in the tendon core during early tendon healing. However, this prediction depended heavily on the hypothesized characteristic that collagen production decreases for supraphysiological levels of mechanical stimulation. A similar idea was implemented in another mechano-regulatory framework [15] where the net collagen production for high strains decreased due to increased collagen degradation and damage. However, the exact mechanobiological mechanisms underlying the decrease in collagen production due to high loading are not well understood. Literature data remains inconclusive on the extent of collagen production during high levels of mechanical stimulation and therefore this study explored different options for collagen production at high strain levels. In addition, we investigated the shape of production law 2. However, all perturbations of production law 2 (S1 Fig) predicted decreased collagen production in the callus core during early tendon healing (S2 Fig) and only minor changes in the temporal evolution of stiffness (S3 Fig).

It is also known that distinct collagen types (type 1 and 3) are observed during tendon healing. Yet, we currently only considered the total collagen phase (not accounting for the different

collagen types) to capture bulk tissue production during healing. This was because important information about the different collagen types during tendon healing are currently not known, e.g. the spatio-temporal distribution of the different collagen types, the mechanical properties of the different collagen types, and the potential differences in the collagen production (and degeneration) mechanisms. However, the different roles and properties of collagen type-1 and 3 will be investigated in future studies. Different levels of strain transfer have been observed in soft tissues, e.g. tendon [55] and meniscus [56]. Similar levels of strain transfer have been identified in in vitro experiments that apply strains to cells on a flexible substrate [57]. Another numerical Achilles tendon healing study implemented a constant strain transfer of 28% between tissue and cell-level strains[14]. However, since a thorough characterization of strain transfer across multiple length scales is lacking, and the available data implies similar strain attenuation in in vitro and in vivo experiments, we did not implement strain transfer from the matrix-level to cell-level strains.

In addition to direct measures of collagen production, experimental studies have investigated how increased levels of loading affect cell behavior in the tendon, which subsequently can affect matrix content. Increased loading has been found to cause onset of tendinopathic cell phenotypes [58–60], non-tenogenic differentiation of tendon stem/progenitor cells into adipocytes, chondrocytes and osteocytes [61,62] and decreased cell-viability [63]. In terms of matrix production, increased loading has increased microdamage-regulated and mechano-regulated gene expression for extracellular matrix production during early tendon healing [8,64]. These cell-related mechanobiological observations may also (in)directly affect the spatio-temporal evolution of collagen content during early tendon healing.

There are also non-mechanobiological factors affecting tendon healing. One of the main reasons why tendons have such a low regenerative potential is the limited intrinsic repair potential of tenocytes in the tendon [1,63]. Other possible (related) causes of delayed production in the tendon core during early tendon healing are decreased oxygen or blood supply [65] combined with the low number of cells in the tendon core [66]. These 'biological factors' will be considered in future works to investigate the contribution of non-mechanobiological mechanisms to tendon healing. For example, recent work by Chen et al. (2018) considered cellular processes, such as cell migration into the healing callus from the intact tendon stumps into the callus core. Recent investigations on spatio-temporal cell migration during tendon healing in flexor tendon [13,67], patellar tendon [16] and Achilles tendon [68] provide valuable experimental data to allow for numerical studies to identify cell- and mechano-regulated mechanisms explaining spatial variations observed during tendon healing.

The implemented reorientation mechanism ensured a homogeneous evolution of collagen alignment that progressed from an initially isotropic towards an aligned collagen network within the first 4 weeks of healing (Figs 4 and S4) [3]. Sensitivity analysis of the rate of reorientation showed only slight changes in the observed trends for the spatio-temporal evolution of the collagen content and temporal evolution of stiffness (S5 Fig). Variations in the callus geometry did not affect the spatio-temporal evolution of collagen alignment (S7 and S10 Figs). Recently, a different numerical implementation of a mechano-regulatory framework also predicted a similar evolution of collagen alignment during early tendon healing [14]. Additional experimental and computational efforts are needed to gain more insight in mechanisms underlying the spatio-temporal evolution of collagen alignment since the few available experimental studies indicate heterogeneous development of collagen alignment during tendon healing [3,11,12].

The different models led to a modest spread in stiffness evolution that was smaller than the standard deviation in the experimental data and captured the experimentally observed stiffness recovery at all time points (1, 2 and 4 weeks). However, we calibrated the initial callus density

(1%) such that the initial stiffness (5% of intact stiffness) after transection at day 1 was based on the earliest available mechanical data [6]. This slight overestimation of the tendon stiffness during the first days of healing could result in underestimated biophysical stimuli in the callus. However, an increase in strains during the first few days of healing would lead to a more pronounced difference between production law 1 and 2 in our models.

In this study, we did not account for changes in the callus geometry during the healing process, although this has been reported in literature (e.g. [6,7,10]). However, we perturbed our callus geometry (increasing cross-sectional area, callus height, overlap of the callus with the intact tendon stumps) to determine the influence of the callus geometry on the spatio-temporal evolution of collagen content for production law 2 (S6, S8 and S9 Figs). We observed that increases in the cross-sectional area (S6 Fig) or the callus height (S9 Fig) did not affect the prediction of delayed and decreased collagen production in the callus core. However, increasing overlap of the callus around the intact stumps predicted more homogeneous collagen production (S7 Fig). Although histological sectioning of healing tendons has provided some insight on the shape of the healing callus, future experimental works could provide more information on the 3D geometry of the healing callus, and other important geometrical features such as the interfaces of the tendon with the intact tendon stumps and the surrounding tissues.

One of the most important factors governing tendon healing is the degree of external loading on the healing tendon. An increasing number of small animal studies have investigated how different durations and extents of loading or immobilization affects the temporal evolution of tendon properties during healing. However, the amount of experimental measurements on spatio-temporal investigations of collagen content and alignment during early tendon healing for different extents of loading is limited to our own recent study [11]. These types of measurements would provide a foundation for numerical frameworks to investigate (mechano-regulatory) principles underlying spatial and temporal variations within tendon healing.

Predicting long-term homeostasis was not within the scope of this paper. Therefore, the model did not incorporate collagen degeneration (e.g. due to proteolytic activity; [15]). Still, the model approached steady-state after 6 weeks of healing as the mean content, alignment and the stiffness at that point changed less than 5% compared to the previous week (S11 Fig). However, the model is flexible and remodeling and degeneration will be addressed in the future.

In summary, we propose a numerical framework to investigate mechano-regulatory processes during tendon healing. An established finite element approach was utilized to simulate normal loading of ruptured rat Achilles tendon to determine the spatial distribution of tensile strain throughout the healing callus. Subsequently, strain-regulated laws predicted spatio-temporal evolution of collagen production and collagen reorientation. Literature inspired a novel strain-dependent collagen production law, which predicted recent experimental findings that collagen production in the tendon core is delayed and decreased throughout the first 2 weeks of tendon healing. Additionally, our computational model captured overall collagen alignment and evolution of stiffness similarly to experimental data. This study is the first numerical modeling approach to investigate mechano-regulated processes underlying spatial and temporal evolution of important tendon properties during healing, and we identified that mechano-regulated collagen production may play a key role in steering the quality of early tendon healing.

## Supporting information

**S1 Table. Summary of the literature data used for designing the strain-dependent collagen production laws.** In vitro strain-stimulation of fibroblasts from various sources. Collagen type

1 and 3 production is expressed as a relative increase to collagen content levels measured without strain stimulation measured after 12–48 hours of strain stimulation.
(DOCX)

**S1 Fig. Variations tested for strain-dependent tissue production law 2.** Three different center transition points (12.5%, 15.0% and 17.5%) and steepness ($k_{sig}$ = 37.5, 75, 150) parameters were tested.
(TIF)

**S2 Fig.** The spatio-temporal evolution of collagen content (% of intact) at week 1, 2 and 4 upon perturbations in the center transition point (low: 12.5%, default: 15.0% and high: 17.5%) and steepness (ksig—low: 37.5, default: 75, high: 150) in production law 2. The width of the stumps is denoted by the black dotted lines. All perturbations of the model parameters predict a decrease in tissue production in the tendon core at week 1.
(TIF)

**S3 Fig. The temporal evolution of tendon stiffness at week 1, 2 and 4 upon perturbations in the center transition point and steepness of production law 2, compared to the experimental data from Khayyeri et al. (2020).** Overall, all different models predicted a development of stiffness in the range of the experimental data. Note that the daily loading simulation for the model with center transition point (12.5%) did not converge beyond 12 days of healing, yet the predicted stiffness was well within the range of experimental data.
(TIF)

**S4 Fig. The temporal evolution of mean tissue alignment (see Eq 11) in the healing tendon callus for the default reorientation rate ($\kappa$ = 0.06, see Eq 10) for the model with production law 2.**
(TIF)

**S5 Fig. Parameter sensitivity of the speed of reorientation (see Eq 10) for the model with production law 2.** Three reorientation speeds ($\kappa$ = 0.03–0.06–0.09) were evaluated, such that the majority of reorientation is completed in 6, 4 and 2 weeks, respectively. For each reorientation rate, the evolution of spatio-temporal evolution of collagen content and temporal evolution of stiffness is shown, all with production law 2. The width of the stumps is denoted by the black dotted lines. All models predicted decreased collagen content in the tendon core at week 1 of healing. Additionally, this effect was more prominent and persistent for the slowly reorienting model ($\kappa$ = 0.03). The stiffness increased with increasing reorientation speed, but the predicted stiffnesses remained within the range of experimental data (Khayyeri et al., 2020).
(TIF)

**S6 Fig. The spatio-temporal evolution of collagen content (% of intact) at week 1, 2 and 4 upon perturbations in cross-sectional area (1X and 2X CSA) and stump overlap (0-50-100%) for the model with production law 2.** When increasing the stump overlap, the collagen production becomes more homogeneous.
(TIF)

**S7 Fig. The spatio-temporal evolution of the average collagen orientation with respect to the longitudinal axis at week 1, 2 and 4 upon perturbations in cross-sectional area (CSA) and stump overlap for the model with production law 2.** The spatio-temporal patterns of reorientation were similar in all models.
(TIF)

**S8 Fig. The spatio-temporal evolution of collagen content (% of callus content) across the midtendon cross section at week 1, 2 and 4 upon perturbations in cross-sectional area (CSA) and stump overlap for the model with production law 2.** The width of the stumps is denoted by the black dotted lines. Decreased content in the tendon core is denoted by a red arrow.
(TIF)

**S9 Fig. The spatio-temporal evolution of collagen content (% of intact) across the midtendon cross section and for the whole callus at week 1, 2 and 4 for the default and 50% longer callus for the model with production law 2.** The width of the stumps is denoted by the black dotted lines. Decreased content in the tendon core is denoted by a red arrow. The model with increased callus height predicted a more homogeneous collagen production.
(TIF)

**S10 Fig. The spatio-temporal evolution of the average collagen orientation at week 1, 2 and 4 for the default and 50% longer callus for the model with production law 2.** The model with increased callus height displayed a very similar evolution of the spatial distribution of collagen alignment.
(TIF)

**S11 Fig. Long-term prediction of collagen content, collagen alignment, overall tendon stiffness and the relative change in these properties for the model with production law 2.** To characterize the long-term predictions of the current healing framework, the model with production law 2 ran for 100 days (~14 weeks). All monitored properties (mean tissue content, alignment, and stiffness) approached steady-state within 7 weeks of healing. The dashed line represents 5% relative change. Within 4 to 7 weeks, all properties changed less than 5% with respect to previous week.
(TIF)

## Acknowledgments

The authors would like to acknowledge the close collaboration with Prof Pernilla Eliasson from Linköping University.

## Author Contributions

**Conceptualization:** Thomas Notermans, Hanifeh Khayyeri, Hanna Isaksson.

**Formal analysis:** Thomas Notermans.

**Funding acquisition:** Hanna Isaksson.

**Investigation:** Thomas Notermans.

**Methodology:** Thomas Notermans, Petri Tanska, Rami K. Korhonen, Hanifeh Khayyeri, Hanna Isaksson.

**Writing – original draft:** Thomas Notermans.

**Writing – review & editing:** Thomas Notermans, Petri Tanska, Rami K. Korhonen, Hanifeh Khayyeri, Hanna Isaksson.

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
