## [Decision Letter · Decision Letter 0]

23 Sep 2020

Dear Mr. Notermans,

Thank you very much for submitting your manuscript "A numerical framework for mechano-regulated tendon repair – simulation of early healing of the Achilles tendon" for consideration at PLOS Computational Biology. I apologize for the delay in review.

As with all papers reviewed by the journal, your manuscript was reviewed by members of the editorial board and by several independent reviewers. In light of the reviews (below this email), we would like to invite the resubmission of a significantly-revised version that takes into account the reviewers' comments.

This manuscript demonstrates novelty and has many merits of interest to the field of tendon biology. The reviewers highlight ways in which the model can be clarified and further justified to improve the quality and potential impact of this work. Namely, including a discussion about the limitations of the model, further justify or revise assumptions, address how temporal changes in tendon stiffness can influence the model, and perform sensitivity analyses related to orientation.

We cannot make any decision about publication until we have seen the revised manuscript and your response to the reviewers' comments. Your revised manuscript is also likely to be sent to reviewers for further evaluation.

Sincerely,

Megan Killian

Guest Editor

PLOS Computational Biology

Mark Alber

Deputy Editor

PLOS Computational Biology

This manuscript demonstrates novelty and has many merits of interest to the field of tendon biology. The reviewers highlight ways in which the model can be clarified and further justified to improve the quality and potential impact of this work. Namely, including a discussion about the limitations of the model, further justify or revise assumptions, address how temporal changes in tendon stiffness can influence the model, and perform sensitivity analyses related to orientation.

Reviewer's Responses to Questions

**Comments to the Authors:**

Reviewer #1: In this study, the authors developed a computational modeling framework to investigate the spatio-temporal patterns of collagen synthesis (and the resulting changes in stiffness) in the healing Achilles tendon. According to their simulations, which utilized two different strain-dependent collagen production laws, spatial variations in strain across the healing tendon led to reduced collagen synthesis in the tendon core during early stages of healing. These findings are consistent with experimental observations. In addition, the model predicted the evolution of tendon stiffness after rupture somewhat accurately.

This well-written study provides a possible explanation for the well-documented spatial heterogeneity of healing tendons and highlights the importance of mechanical loading in tendon regeneration. Thus, the findings of this study are of interest to the field. However, some key details about the model are missing, some important sensitivity analyses have not been performed and whether the model reflects a realistic mechanobiological response is not clear. Specifically, the following concerns should be addressed by the authors:

1) According to both production laws used by the authors, the collagen production rate is always greater than (or equal to) zero. Thus, a cyclically loaded tendon never reaches an equilibrium “attractor state” where (net) collagen production ceases. That is, in their model, a cyclically loaded (healing) tendon will continue to produce collagen and stiffen indefinitely. This behavior is clearly not physically realistic. Other models typically allow for reduction of collagen content due to proteolytic activity for certain levels of strain (e.g., when strain levels are below a threshold). In this way, the material properties adapt until equilibration when the optimal strain state is reached. Can the authors comment on this key limitation of their model? Why did they choose to not allow for decreased collagen content under some conditions?

2) The authors perform a sensitivity analysis to determine how changes in production law 2 influence collagen content across the callus (Figure S2). However, it would be helpful if they also show how these changes influence the temporal evolution of tendon stiffness. If the production law is altered, are the time-dependent changes in tendon stiffness still consistent with experimental findings?

3) It is described that the reorientation rate was chosen to match findings in the literature. But there is no sensitivity analysis performed to determine how changes in reorientation rate influence key output parameters.

4) It appears that the model includes two forms of adaptation: strain-regulated collagen synthesis and collagen reorientation that in turn alters the mechanical response of the healing tendon through the fiber-reinforced model. It would be helpful and interesting if the authors delineate the importance of each of these forms of adaptation by performing simulations (a) with only strain-regulated collagen synthesis and (b) with only collagen reorientation. Does the model’s agreement with experimental data depend on “turning on” both of these forms of adaptation?

5) In many of the studies cited to justify the strain-dependence of collagen production, strains were directly applied to cells on a flexible substrate. In healing tendons (as in other fibrous tissues), tissue- and matrix-scale strains are not equal to strains experienced by resident cells. Instead, tissue- and matrix-scale strains are often amplified or attenuated across length scales (see, for example, PMID 26726994) . Thus, it may not be appropriate to rationalize tissue strain-dependent collagen production in healing tendons based on data from directly deformed cells.

6) Was a convergence analysis performed to ensure that the mesh was sufficiently dense?

7) The authors should include the constitutive equation that they used in the main manuscript rather than simply providing a reference.

Reviewer #2: PCOMPBIOL-D-20-01279

"A numerical framework for mechano-regulated tendon repair – simulation of early healing of the Achilles tendon".

This interesting study proposes a mechano-regulatory framework to describe spatio-temporal evolution of collagen content and alignment and the temporal evolution of stiffness during tendon healing. The novel methodology includes strain-regulated laws to compare tendon callus properties in the core and peripheral regions. Collagen production in the tendon core was delayed and decreased throughout the first 2 weeks following tendon injury. Favorable comparisons of the model outputs to experimental data indicate preliminary reliability of this model.

The novelty of this work regards the spatial and temporal descriptions of callus deformation and the consideration of supraphysiologic levels of strain. While it is fully understood that the reliability of a numerical simulation is gauged from comparison of its predictions to available experimental results, it is felt that this model provides only a modest advancement. Specifically, the model considers only the callus, with little detail paid to the boundary conditions (e.g., adjoining tendon stumps) aside from the callus-stump interfaces. The model also relies on a large number of assumptions (e.g., apparently no distinction between the fibrillar collagens) and the loading conditions (lines 150-153) are not well justified.

Specific Comments:

ABSTRACT

Lines 35 -38: restate this sentence to convey that your results (biomechanical and collagen orientation) suggest improved healing

INTRODUCTION

General comment:

By “tendon repair,” are the authors alluding to full width transection models with or without surgical repair?

Lines 71-72: for reference 11, is the structural analysis based on collagen fibers or fibrils, or both?

Line 72: consider removing “see” from (see Fig 1)

Lines 78-79: consider replacing “histology work” with “histologic analysis.” What type of collagen (I, III) staining are you referring to?

Line 89: remove “Only” from “only few”

Line 104: it is assumed that collagen production (which of course is a central focus of the current study) refers to biochemical compositional measures of collagen and not collagen gene expression. Furthermore, is it an accurate statement that the authors are not distinguishing collagen I from collagen III (or other fibrillar collagens)? Col III matrix tensile properties differ from those of Col I. Is it necessary for the model to account for the gradual temporal reduction in Collagen type III:I ratio that presumably occurs during the healing process? Please clarify these points.

Line 135: please provide a rationale for the prescribed zero pore pressure.

Lines 145-146: is sigma in these equations a generic variable for tendon properties (many readers may associate sigma with mechanical stress)

Lines 147-9: please clarify this approach. It is unclear (to this reviewer) how a callus density of 1% of the intact tendon matches the d3 tendon stiffness. The cross-sectional area presumably is also altered (and modeled) from experimental data? What is the normalized stiffness at d3 post-injury? Please provide a justification for the selection of 13 random initial fiber directions.

Lines 150-53: how sensitive is the model output to these tendon loading rates? Does the model only consider the peak daily load (as opposed to the # of loading cycles) which is the mechanical stimulus for collagen remodeling/production?

Adaptive mechanobiological model: the authors detail the callus matrix mechanical behavior, but the boundary conditions are minimally addressed. What initial stiffness values are assigned to the adjoining stumps, and do these change over time (post-injury)? Do the intact tendon stumps strain-shield the callus? At strain levels exceeding 15%, it can be argued that collagen fibers may undergo plastic deformation or damage, thereby altering their orientation and load-bearing capacity. Does the model account for the latter occurrences?

Lines 199-200: please provide references for the “well established alignment criterion”

RESULTS

Can the model predict callus rupture strain, load, and location? I would expect that most of these experimental data are also available from prior studies. If prior studies reported optical strain during tensile loading, it would be interesting to know whether the core vs peripheral regions exhibited differential strain responses at the different time points.

Figure 6: please clarify that the temporal nature of collagen content changes for law 1 was limited to the core region (as Figure C does not include a ‘2’ for law 1)

DISCUSSION

This reviewer understands and appreciates the authors’ assumption that healing is a strain-driven process (strain-dependent production laws), but I think that the complex interrelationships between collagen content, fiber orientation, stiffness and principal strain warrants additional discussion.

For example, please consider that collagen production and orientation could be influencing the strain and stiffness behavior.

Lines 320-321: If my understanding is correct, please state more clearly that the non-collagenous tissue composition is not considered in this model.

**Have all data underlying the figures and results presented in the manuscript been provided?**

Reviewer #1: **No: **The authors did not provide their raw data, but stated that it is available by request.

Reviewer #2: Yes

PLOS authors have the option to publish the peer review history of their article (what does this mean?). If published, this will include your full peer review and any attached files.

Reviewer #1: No

Reviewer #2: No
---

## [Decision Letter · Decision Letter 1]

15 Dec 2020

Dear Mr. Notermans,

We are pleased to inform you that your manuscript 'A numerical framework for mechano-regulated tendon healing– simulation of early regeneration of the Achilles tendon' has been provisionally accepted for publication in PLOS Computational Biology.

Before your manuscript can be formally accepted you will need to complete some formatting changes, which you will receive in a follow up email. A member of our team will be in touch with a set of requests. Additionally, please address the following:

-- Add a statement clarifying if you considered variable tissue permeability for your poroviscoelastic portion of your model.

-- In the supplement or for Figure 4, include the original (empirical) data from the literature or explain why these data were reproduced with 'arbitrary units.'

Best regards,

Megan Killian

Guest Editor

PLOS Computational Biology

Mark Alber

Deputy Editor

PLOS Computational Biology

Reviewer's Responses to Questions

**Comments to the Authors:**

Reviewer #1: Thank you for the very detailed and thorough responses to the reviewer comments. I have no further concerns.

Reviewer #2: Thanks to the authors for their detailed clarifications, which have adequately addressed my concerns.

Reviewer #3: Interesting paper regarding a computational model to understanding collagen synthesis and organization during Achilles tendon healing. I have some minor questions regarding model assumptions that should be addressed in the text:

1. Is fibril size (diameter or length) considered? This parameter could have a significant effect on mechanoregulation as strains would vary based on fibril size. This is particularly relevant for healing as microfibrils are first deposited and then fibril growth via molecular accretion and fibril fusion are occuring simultaneously. In addition, this process (fibril deposition and growth) are also mechanosensitive.

2. This model assumes that ground substance is produced at a similar rate to the collagenous matrix. However, we know that turnover of the non-collagenous matrix occurs must faster than turnover of collagen. How do you account for this? Have you performed any analyses considering dissimilar production? This will surely affect the mechanical properties of the tissue as well as the maximum principal strains on the tissue.

3. Did you consider variable tissue permeability for your poroviscoelastic portion? Collagen structure, particularly collagen fibril packing, as well as the non-collagenous matrix composition will influence poroelasticity of the tissue and it's likely that this parameter changes over the course of healing.

4. Figure 4 - Data from the literature is reproduced with 'arbitrary units'. What's the purpose behind this? Is it possible to put the real data on a separate axis with actual units? This would make the figure much more believable. If not, how did you plot this data?

**Have all data underlying the figures and results presented in the manuscript been provided?**

Reviewer #1: **No: **The authors have only stated that the raw data will be provided upon request.

Reviewer #2: Yes

Reviewer #3: **No: **Authors say raw data available upon request.

PLOS authors have the option to publish the peer review history of their article (what does this mean?). If published, this will include your full peer review and any attached files.

Reviewer #1: No

Reviewer #2: No

Reviewer #3: No

---

## [Editor Report · Acceptance letter]

27 Jan 2021

PCOMPBIOL-D-20-01279R1 

A numerical framework for mechano-regulated tendon healing – simulation of early regeneration of the Achilles tendon

Dear Dr Notermans,

I am pleased to inform you that your manuscript has been formally accepted for publication in PLOS Computational Biology. Your manuscript is now with our production department and you will be notified of the publication date in due course.

With kind regards,

Alice Ellingham
